# LL-37 and Double-Stranded RNA Synergistically Upregulate Bronchial Epithelial TLR3 Involving Enhanced Import of Double-Stranded RNA and Downstream TLR3 Signaling

**DOI:** 10.3390/biomedicines10020492

**Published:** 2022-02-19

**Authors:** Sara Bodahl, Samuel Cerps, Lena Uller, Bengt-Olof Nilsson

**Affiliations:** Department of Experimental Medical Science, Lund University, BMC D12, SE-22184 Lund, Sweden; sara.bodahl@med.lu.se (S.B.); samuel.cerps@med.lu.se (S.C.); lena.uller@med.lu.se (L.U.)

**Keywords:** host defense peptide, innate immunity, NF-κB, poly I:C, toll-like receptor 3

## Abstract

The human host defense peptide LL-37 influences double-stranded RNA signaling, but this process is not well understood. Here, we investigate synergistic actions of LL-37 and synthetic double-stranded RNA (poly I:C) on toll-like receptor 3 (TLR3) expression and signaling, and examine underlying mechanisms. In bronchial epithelial BEAS-2B cells, LL-37 potentiated poly I:C-induced TLR3 mRNA and protein expression demonstrated by qPCR and Western blot, respectively. Interestingly, these effects were associated with increased uptake of rhodamine-tagged poly I:C visualized by immunocytochemistry. The LL-37/poly I:C-induced upregulation of TLR3 mRNA expression was prevented by the endosomal acidification inhibitor chloroquine, indicating involvement of downstream TLR3 signaling. The glucocorticoid dexamethasone reduced LL-37/poly I:C-induced TLR3 expression on both mRNA and protein levels, and this effect was associated with increased IκBα protein expression, suggesting that dexamethasone acts via attenuation of NF-κB activity. We conclude that LL-37 potentiates poly I:C-induced upregulation of TLR3 through a mechanism that may involve enhanced import of poly I:C and that LL-37/poly I:C-induced TLR3 expression is associated with downstream TLR3 signaling and sensitive to inhibition of NF-κB activity.

## 1. Introduction

Toll-like receptors (TLRs) are important innate immune receptors that trigger defense and inflammatory responses in host cells in response to various microorganisms and pro-inflammatory agents [1]. The TLR family consists of about ten members of plasma membrane and intracellular receptors in humans, and their expression is largely dependent on cell type [2]. TLR3 is intracellularly expressed and known to recognize and respond to double-stranded (ds) RNA, usually from viral origin. The activated TLR3 receptor increases expression of transcription factors interferon regulatory factor 3 and 7, but also nuclear factor kappa B (NF-κB), leading to the production and secretion of type I interferons and pro-inflammatory cytokines [3,4,5,6].

The human host defense peptide LL-37 is part of the first line of defense against invading pathogens [7]. The peptide is mainly produced by neutrophils, but also by other types of white blood cells, skin epithelial cells and epithelial cells aligning the mucosal areas including bronchial epithelial cells [8,9,10]. LL-37 exerts antimicrobial activity through permeabilization of the bacterial cell wall, resulting in cell lysis, but also via neutralization of bacterial lipopolysaccharides [11,12]. Besides showing activity against microorganisms, LL-37 also modulates the immune system by acting as a chemoattractant for several types of immune cells and by promoting their expression of pro-inflammatory cytokines [13,14,15,16]. High concentrations (>1 µM) of LL-37 have been found locally in the gingival crevicular fluid collected at disease sites of patients suffering from periodontitis and in psoriatic skin lesions [17,18]. After allergen challenge, LL-37 appears on the bronchial epithelial surface as a component of plasma exudation, which is an early defense/inflammatory response in health and disease [19]. Bals et al. [20] show that LL-37 is expressed in surface epithelia in healthy human lung tissue, and moreover these authors report that the peptide shows activity against *Pseudomonas aeruginosa*, a pathogen associated with pneumonia. Thus, it is of in vivo relevance, to investigate the functional importance of LL-37 in lung epithelial cells.

Polycytidylic acid (poly I:C) is a synthetic analog to dsRNA, and a well-studied TLR3 ligand [5,21]. LL-37 has been shown to potentiate poly I:C-induced production of pro-inflammatory cytokines in human airway epithelial cells, and this effect is supposed to involve an LL-37-induced increase in the intracellular bioavailability of poly I:C [22,23]. We have recently demonstrated that LL-37 facilitates the pro-inflammatory effects of poly I:C in human coronary artery smooth muscle cells by upregulation of TLR3 expression, suggesting that LL-37 can promote poly I:C-induced inflammation also via this mechanism [24]. However, the mechanism explaining how LL-37 can cause upregulation of TLR3 expression in the presence of poly I:C has not been identified.

In the present study, we investigate the mechanisms behind LL-37/poly I:C-induced upregulation of TLR3 in the human bronchial epithelial BEAS-2B cell line. We show that LL-37 potentiates poly I:C-induced upregulation of TLR3 expression, and that this effect is associated with activation of TLR3 signaling and sensitive to pharmacological inhibition of NF-κB activity. Moreover, we demonstrate that LL-37 enhances uptake of poly I:C, suggesting that LL-37 may potentiate poly I:C-induced TLR3 expression via this mechanism.

## 2. Materials and Methods 

### 2.1. Cells and Cell Culture

The human bronchial epithelial cell line BEAS-2B cells were purchased from ATCC, and cultured in RPMI 1640 medium supplemented with Glutamax, (Thermo Fisher Scientific, Waltham, MA, USA), fetal bovine serum (10%, Thermo Fisher Scientific), penicillin (50 U/mL) and streptomycin (50 μg/mL). The BEAS-2B cells were cultured under standard conditions in a cell incubator, reseeded upon reaching confluence as appropriate and counted (LUNA automated cell counter, Logos Biosystems, Anyang-si, Korea) to ensure suitable cell density. All experiments were performed in the culture medium specified above.

### 2.2. Real-Time RT-qPCR

Total RNA was extracted from cell lysates and purified with a miRNeasy kit (Qiagen, Venlo, Netherlands), using the QIAcube instrument (Qiagen) according to manufacturer’s instructions. Concentration and quality of the RNA was determined by a NanoDrop 2000C spectrophotometer (Thermo Fisher Scientific). The RNA was analyzed in a Step-One Plus real-time cycler (Applied Biosystems, Waltham, MA, USA), using a QuantiFast SYBR green RT-PCR kit (Qiagen), according to the manufacturer’s guidelines. QuantiTect primer assays (Qiagen) were used for TLR3 (Hs_TLR3_1_SG) and GAPDH (Hs_GAPDH_2_SG), with GAPDH serving as an internal control. All samples were analyzed in duplicate and gene expression was calculated using the delta CT method as previously described [25]. 

### 2.3. Western Blot

Cells were lysed in sodium dodecyl sulfate (SDS) buffer and total protein concentration in each sample determined (Bio-Rad DC protein assay, Bio-Rad, Hercules, CA, USA). Identical amounts of total protein were loaded to each lane on Criterion TGX 4–15% precast gels (Bio-Rad) and proteins separated using SDS/PAGE. The proteins were then transferred to nitrocellulose membranes (Trans-Blot Transfer system, Bio-Rad). Membranes were blocked in Tris-buffered saline (TBS, Bio-Rad) with 0.5% casein and incubated 12–15 h (4 °C) with primary TLR3 (Cell Signaling, #6961S, rabbit) or NF-κB subunits phosphorylated NF-κB p65 (Cell Signaling, #3031S, rabbit), phosphorylated NF-κB p105 (Cell signaling, #4806S, rabbit) and IκBα (Cell Signaling, #9242S, rabbit) antibodies. The GAPDH protein (Merck Millipore, Burlington, MA, USA, #MAB374, mouse), was used in all blots as an internal control. Membranes were then incubated with horse-radish peroxidase conjugated secondary antibodies (Cell Signaling, #7074S, rabbit or #7076S, mouse) for 2 h at room temperature. TBS-Tween(T) was used to wash the membranes between each incubation step and both primary and secondary antibodies were diluted in 0.5% casein/TBS-T as appropriate. SuperSignal West Femto chemiluminescence reagent (Thermo Fisher Scientific) was used to visualize the immunoreactive bands, using a LI-COR Odyssey Fc instrument (LI-COR Biosciences, Lincoln, NE, USA).

### 2.4. Assessment of Cell Viability 

Cell viability was assessed using the thiazolyl blue tetrazolium (MTT, Sigma-Aldrich, St Louis, MO, USA) assay. Cells were seeded in a 96-well plate and incubated with MTT solution (0.5 mg/mL, Sigma-Aldrich) for 1 h at 37 °C. The supernatants were then discarded, and the formazan product dissolved in dimethyl sulfoxide (DMSO). The plate was analyzed in a Multiscan GO Microplate Spectrophotometer (Thermo Fisher Scientific), measuring the absorbance at 540 nm. 

### 2.5. Measurement of Poly I:C Import Using Fluorescence Imaging

Cells were grown on round glass coverslips and placed in a 24-well cell culture plate. The cells were treated with rhodamine-labeled poly I:C (InvivoGen), either alone or in combination with LL-37 for 6 and 24 h at 37 °C. The cells were fixed with 4% paraformaldehyde for 10 min and the coverslips mounted with Fluoroshield mounting media (Sigma-Aldrich) containing the nuclear marker DAPI. Phosphate buffered saline (PBS, Gibco, Waltham, MA, USA) was used to wash the cells between each incubation step. The cells were photographed and analyzed in a fluorescence microscope (Olympus, Olympus Europa, Hamburg, Germany). Quantitative assessment was performed in the Image J software. For each coverslip, average cellular fluorescence intensity was calculated by analyzing the fluorescence signal from five cells in three different areas. 

### 2.6. Agents

LL-37 (Bachem, Bubendorf, Switzerland) and dexamethasone (Sigma-Aldrich) were dissolved in DMSO. Poly I:C (InvivoGen, San Diego, CA, USA), fluorescent rhodamine-tagged poly I:C (InvivoGen) and chloroquine (Sigma-Aldrich) were dissolved in PBS. Vehicle was added as appropriate. Cells were pre-treated with dexamethasone or chloroquine for 30 min and dexamethasone or chloroquine were then present throughout the experiment. 

### 2.7. Statistics

Data were analyzed in GraphPad Prism9 (GraphPad Software) and presented as mean ± SEM. Each experiment was repeated 2–4 times. Each culture well was regarded to represent one biological replicate (*n* = 1), except for assessment of poly I:C import using cells grown on glass coverslips where each independent experiment was regarded to represent one biological replicate (*n* = 1). The n-values for each experiment are displayed in the figure legends. A one-way ANOVA, followed by Tukey’s multiple comparison post hoc tests was used to calculate statistical significance as appropriate. *p* value less than 0.05 were considered statistically significant.

## 3. Results

### 3.1. Dexamethasone Reduces TLR3 Expression Induced by Combined Treatment with LL-37 and Poly I:C in BEAS-2B Cells

Initially, we investigated TLR3 mRNA expression in human bronchial epithelial BEAS-2B cells treated with poly I:C (10 µg/mL) alone or LL-37 (1 µM) and poly I:C (10 µg/mL) in combination (LL-37/poly I:C), in the presence or absence of dexamethasone (1 µM) for 6 and 24 h. We have previously demonstrated that combined treatment with LL-37 and poly I:C for 24 h triggers production of pro-inflammatory IL-6 and MCP-1 in human coronary artery smooth muscle cells, suggesting that stimulation with LL-37 and poly I:C in combinations is pro-inflammatory at this time point [24]. Hence, it is reasonable to assume that treatment with LL-37/poly I:C will affect TLR3 expression within 24 h. Both treatment with poly I:C alone and LL-37/poly I:C for 6 h increased the TLR3 transcript levels compared to control cells, though the co-treatment had a stronger effect compared to treatment with poly I:C alone (Figure 1A). Dexamethasone reduced the poly I:C-induced TLR3 mRNA expression at 6 h by 35%, whereas it reduced the LL-37/poly I:C-induced transcript levels by 60% (Figure 1A). Both treatment with poly I:C alone and LL-37/poly I:C for a longer time (24 h) increased the TLR3 mRNA levels compared to control cells (Figure 1B). Dexamethasone reduced poly I:C- and LL-37/poly I:C-induced TLR3 mRNA expression at 24 h by 45 and 60%, respectively (Figure 1B). Next, we used Western blot analysis to examine the TLR3 protein expression. Here we choose to focus on the co-treatment with LL-37 and poly I:C since this treatment caused an overall larger increase in TLR3 mRNA expression than poly I:C alone. Treatment with LL-37 (1 µM)/poly I:C (10 µg/mL) for 24 h increased the TLR3 protein expression by ten times compared to control cells (Figure 1C). Notably, dexamethasone (1 µM) reduced this effect by more than 60% (Figure 1C). 

### 3.2. Dexamethasone-Induced Down-Regulation of TLR3 Expression Is Associated with Reduced NF-κB Activity in BEAS-2B Cells

To investigate if the transcription factor NF-κB is involved in the LL-37/poly I:C-induced modulation of TLR3 expression, we evaluated the protein expression of NF-κB inhibitor IκBα, NF-κB phosphorylated p65 and NF-κB phosphorylated p105 using Western blot. Cells were treated with or without LL-37 (1 µM)/poly I:C (10 µg/mL) for 24 h in the presence or absence of dexamethasone (1 µM). Both treatment with dexamethasone alone and in combination with LL-37/poly I:C increased the expression of IκBα, suggesting that dexamethasone reduces NF-κB via this mechanism (Figure 2A). The protein expression of NF-κB phosphorylated p65 was not affected by any treatment (Figure 2B). Interestingly, stimulation with LL-37/poly I:C increased the expression of NF-κB phosphorylated p105 compared to control cells, and moreover dexamethasone tended (not statistically significant) to reduce this effect (Figure 2C). Considering that stimulation with LL-37/poly I:C increases phosphorylated p105 but has no effect on either IκBα or phosphorylated p65 levels, it is reasonable to conclude that treatment with LL-37 and poly I:C in combination enhances NF-κB activity. 

### 3.3. Upregulation of TLR3 Expression by Poly I:C and LL-37/poly I:C Involves Downstream TLR3 Signaling in BEAS-2B Cells

In the next set of experiments, we further examined LL-37-induced potentiation of poly I:C-stimulated TLR3 mRNA expression to assess its dependence on poly I:C concentration. For these experiments, we stimulated cells with or without poly I:C at low concentrations (0.2 or 2 µg/mL), LL-37 (1 µM) or the two in combination for 6 and 24 h. Treatment with poly I:C alone increased the TLR3 mRNA expression in a concentration dependent manner at both 6 and 24 h (*p* < 0.01 and *p* < 0.001 for 0.2 vs. 2 µg/mL at 6 and 24 h, respectively) (Figure 3A,B). Interestingly, co-treatment with LL-37 and poly I:C for 6 h increased the TLR3 mRNA levels by 2–3 times, compared to treatment with poly I:C alone, whereas LL-37 did not potentiate the poly I:C-induced TLR3 expression at 24 h, indicating that LL-37 has a rapid turnover (Figure 3A,B). Importantly, LL-37 alone had no effect on the mRNA expression of TLR3 (Figure 3A,B). It has previously been shown that TLR3 signaling depends on endosomal acidification [26], and to assess if LL-37/poly IC-induced upregulation of TLR3 expression is associated with activation of TLR3 signaling, we used the endosomal acidification inhibitor chloroquine. Cells were treated with or without poly I:C (0.2 µg/mL), LL-37 (1 µM) or the two in combination for 6 h, in the absence or presence of chloroquine (2 µg/mL). Treatment with chloroquine completely prevented poly I:C- and LL-37/poly I:C-induced TLR3 mRNA expression (Figure 3C). Chloroquine alone did not affect the TLR3 expression (Figure 3C). 

### 3.4. High but Not Low Concentrations of LL-37 Enhance Poly I:C-Induced TLR3 Expression at 24 h, Indicating Rapid Turnover of LL-37 in BEAS-2B Cells

Since LL-37 (1 µM) potentiates poly I:C-evoked upregulation of TLR3 at 6 but not 24 h, we hypothesize that LL-37 shows rapid turnover. To test this, we examined if treatment with a high (4 µM) but not to a low (1 µM) dose of LL-37 potentiates poly I:C-induced TLR3 expression at 24 h. Interestingly, the combined stimulation with 4 µM LL-37 and 0.2 µg/mL poly I:C enhanced TLR3 mRNA expression by about five times compared to stimulation with poly I:C alone, while the lower concentration of LL-37 (1 µM) caused no potentiation of poly I:C-induced TLR3 expression (Figure 4A). Treatment with LL-37 alone (4 µM) showed no effect on mRNA levels for TLR3 (Figure 4A). Additionally, we also investigated TLR3 protein expression using Western blot analysis. The combined treatment with 4 µM LL-37 and 0.2 µg/mL poly I:C for 24 h augmented TLR3 protein expression by more than three times compared to treatment with poly I:C alone (Figure 4B). High concentrations of LL-37 can be cytotoxic for different types of human cells [27]. To exclude that LL-37 reduces cell viability at the concentrations used here (1 and 4 µM), we assessed viability of BEAS-2B cells using the MTT assay. LL-37 is considered to cause a rapid permeabilization of the plasma membrane, and this effect is associated with reduced cell viability monitored with the MTT method [27]. Treatment with 10 µM LL-37 for 4 h reduced cell viability by around 50%, whereas lower concentrations (0.1, 1 and 4 µM) had no effect (Figure 4C).

### 3.5. LL-37 Increases Import of Poly I:C in BEAS-2B Cells

We hypothesized that potentiation of poly I:C-induced TLR3 expression by LL-37 can be due to LL-37-stimulated import of poly I:C. To address this issue, we treated cells with rhodamine-tagged poly I:C (4 µg/mL), either alone or in combination with LL-37 (4 µM) for 6 and 24 h. A fluorescence signal (red) was observed both in cells treated with poly I:C alone and in cells co-treated with LL-37 and poly I:C, whereas no or very low fluorescence was detected in untreated control cells (Figure 5A–D). Interestingly, fluorescence was around three times stronger in cells treated with LL-37/poly I:C, compared to cells treated with poly I:C alone at both 6 and 24 h, suggesting that LL-37 increases import of poly I:C (Figure 5A–D). 

## 4. Discussion

In the present study, we demonstrate on both transcript and protein levels that the host defense peptide LL-37 potentiates poly I:C-induced upregulation of TLR3 expression in human bronchial BEAS-2B epithelial cells, and that this effect is associated with LL-37-evoked increase of cellular uptake of poly I:C. LL-37-induced stimulation of poly I:C import is observed already at 6 h of co-treatment with LL-37 and poly I:C, suggesting that LL-37 triggers uptake of poly I:C through a rapid process. Previously, Singh et al. [23] have reported that LL-37 promotes poly I:C-induced stimulation of pro-inflammatory cytokine production via a mechanism involving endosomal acidification, leading to increased intracellular bioavailability of poly I:C in the same cell type as used by us here, i.e., BEAS-2B. Hence, it seems that LL-37 facilitates poly I:C signaling through different mechanisms involving both enhanced uptake as demonstrated in the present study and increased processing of endosomal poly I:C. LL-37 has been shown to be internalized by human macrophages and osteoblasts via endocytosis, but inhibition of the endocytic pathways does not completely prevent import of LL-37, suggesting that also other mechanisms besides endocytosis are involved [28,29]. It is well-recognized that LL-37 forms pores in the plasma membrane, and it is plausible that LL-37/poly I:C complexes can use the LL-37 self-made pores to cross plasma membranes [27,30].

Here we show that treatment with 1 µM LL-37 for 6 h stimulates poly I:C-induced upregulation of TLR3 expression, whereas this effect is absent at 24 h, indicating a rapid turnover of the peptide. Interestingly, the peptide can also elevate poly I:C-induced upregulation of TLR3 at 24 h if the cells are treated with a higher concentration (4 µM) of LL-37, suggesting 4 µM LL-37 is necessary to maintain a high enough concentration of LL-37 over the whole 24 h period to potentiate the effect of poly I:C. Indeed, LL-37 has been reported to have a short half-life (~1 h) in cells [23]. In high concentrations (>4 µM), LL-37 may show cytotoxicity by promoting apoptosis in human host cells [27,31]. Importantly, we demonstrate that LL-37 does not reduce BEAS-2B cell viability in the concentrations (1 or 4 µM) used in the present study, arguing that our present results are not influenced by LL-37-induced cytotoxicity.

In the present study, we demonstrate that chloroquine, an inhibitor of endosomal acidification, attenuates poly I:C- and LL-37/poly I:C-induced upregulation of TLR3, suggesting that poly I:C and LL-37/poly I:C driven stimulation of TLR3 expression involves downstream TLR3 signaling. Stimulation of TLR3 expression by LL-37/poly I:C correlates with increased levels of NF-κB phosphorylated p105, suggesting that LL-37/poly I:C-induced signaling downstream of TLR3 involves activation of NF-κB. We show that the NF-κB inhibitor dexamethasone [32], reduces LL-37/poly I:C-evoked upregulation of both TLR3 mRNA expression and protein production and tend to reduce (not statistically significant) LL-37/poly I:C-induced enhancement of NF-κB phosphorylated p105. Importantly, our data show that dexamethasone strongly elevates IκBα levels. Taken together these data provide evidence that LL-37/poly I:C-induced upregulation of TLR3 involves activation of NF-κB. Interestingly, dexamethasone reduces mortality in critically ill COVID-19 patients [33]. The life-threatening cytokine storm induced by coronavirus seems to involve NF-κB activation, and inhibition of NF-κB activity by dexamethasone is an important therapeutic strategy in seriously ill COVID-19 patients [34,35]. The present data support the possibility that dexamethasone antagonizes the cytokine storm in COVID-19 patients in part by reducing expression of the virus signaling associated receptor TLR3.

Respiratory viral infection of human airways involves plasma exudation that, rather than local cells, likely is responsible for appearance of LL-37 in airway surface liquids. Hence, there is opportunity for interactions between viral dsRNA and LL-37. This interaction may be particularly pronounced in bronchial asthma where plasma exudation is exaggerated at viral infection, the latter being the most common cause of asthma exacerbations. The possibility arises that LL-37/dsRNA synergy is involved in TLR3-dependent overproduction of epithelial cytokines of importance in exacerbations asthma, Hence, it is warranted to further explore the possibility that the present synergy between dsRNA and LL-37 is partly involved in epithelium-driven worsening of asthma.

In summary, we conclude that LL-37 potentiates poly I:C-induced upregulation of TLR3 through a mechanism that may involve enhanced import of poly I:C in bronchial epithelial BEAS-2B cells. Furthermore, we demonstrate that LL-37 and poly I:C, acting in synergy, increases TLR3 expression through downstream TLR3 signaling and that this process is sensitive to inhibition of NF-κB activity.

## Figures and Tables

**Figure 1 biomedicines-10-00492-f001:**
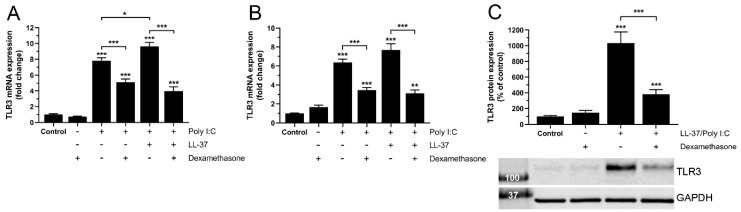
LL-37/poly I:C-induced stimulation of TLR3 expression is reduced by dexamethasone in BEAS-2B cells. (**A**,**B**) TLR3 mRNA expression was determined using quantitative real-time RT-PCR in cells treated with or without poly I:C (10 µg/mL) alone, or LL-37 (1 µM) and poly I:C (10 µg/mL) in combination in the presence or absence of dexamethasone (1 µM) for 6 h (**A**) and 24 h (**B**). (**C**) TLR3 protein expression was assessed using Western blot in cells treated with or without LL-37 (1 µM) and poly I:C (10 µg/mL) in combination (LL-37/poly I:C) in the presence or absence of dexamethasone (1 µM) for 24 h. The TLR3 immunoreactive band was observed at the expected molecular weight of 115–130 kDa and normalized to GAPDH (37 kDa), serving as internal control. (**A**–**C**) Data are presented as mean ± SEM, *n* = 7–8 (**A**,**B**) and *n* = 6 (**C**) in each group. Statistical significance was calculated using a one-way ANOVA, followed by Tukey’s post hoc test. ** and *** represent *p* < 0.01 and *p* < 0.001, respectively, vs. control. For comparisons indicated by the bars, * and *** represent *p* < 0.05 and *p* < 0.001, respectively.

**Figure 2 biomedicines-10-00492-f002:**
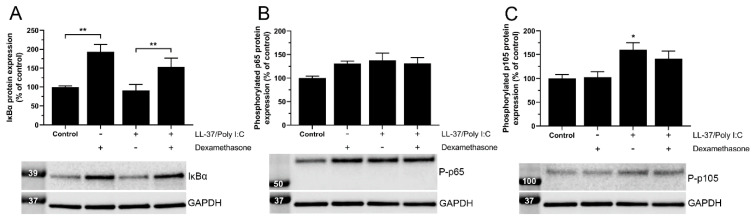
Dexamethasone reduces NF-κB activity in BEAS-2B cells by increasing expression of IκBα. (**A**–**C**) Western blot analysis of IκBα (**A**), NF-κB phosphorylated p65 (**B**) and NF-κB phosphorylated p105 (**C**) was performed in cells stimulated with or without poly I:C (10 µg/mL) and LL-37 (1 µM) in combination in the presence or absence of dexamethasone (1 µM) for 24 h. The immunoreactive bands were observed at the expected molecular weight of 39 kDa (IκBα), 65 kDa (phosphorylated p65) and 105 kDa (phosphorylated p105), and the intensity of each band was normalized to GAPDH (37 kDa), serving as internal control. Data are presented as mean ± SEM, *n* = 7–8 in each group. Statistical significance was calculated using a one-way ANOVA followed by Tukey’s post hoc test. * represents *p* < 0.05 vs. control. For comparisons indicated by the bars, ** represents *p* < 0.01.

**Figure 3 biomedicines-10-00492-f003:**
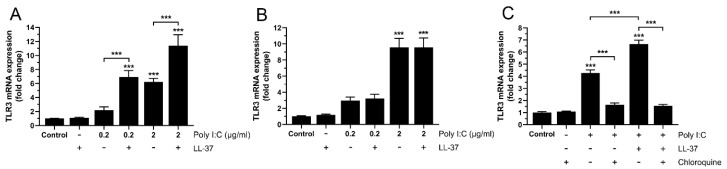
LL-37/poly I:C-induced stimulation of TLR3 mRNA expression is abolished by chloroquine in BEAS-2B cells. (**A**,**B**) TLR3 mRNA expression was analyzed using quantitative real-time RT-PCR in cells treated with or without poly I:C (0.2 or 2 µg/mL) alone, LL-37 (1 µM) alone, or the two in combination for 6 (**A**) and 24 h (**B**). (**C**) TLR3 mRNA expression was determined in cells treated with poly I:C (0.2 µg/mL), LL-37 (1 µM), or the two in combination in the presence or absence of chloroquine (2 µg/mL) for 6 h. (**A**–**C**) Data are presented as mean ± SEM, *n* = 8–12 (**A**), *n* = 6–8 (**B**) and *n* = 4 (**C**) in each group. Statistical significance was calculated using a one-way ANOVA, followed by Tukey’s post hoc test. *** represents *p* < 0.001 vs. control or *p* < 0.001 for comparisons indicated by the bars.

**Figure 4 biomedicines-10-00492-f004:**
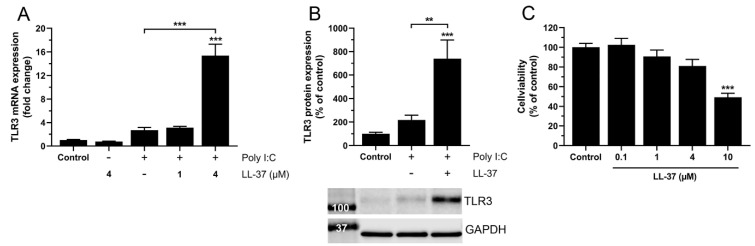
High but not low concentration of LL-37 stimulates poly I:C-induced upregulation of TLR3 in BEAS-2B cells. (**A**) TLR3 mRNA expression was evaluated using quantitative real-time RT-qPCR in cells treated with poly I:C (0.2 µg/mL), LL-37 (4 µM) or LL-37 (1 and 4 µM) and poly I:C (0.2 µg/mL) in combination for 24 h. (**B**) TLR3 protein expression was evaluated using Western blot in cells treated with poly I:C (0.2 µg/mL) alone or poly I:C in combination with LL-37 (4 µM) for 24 h. The TLR3 immunoreactive band was observed at the expected molecular weight of 115-130 kDa and normalized to GAPDH (37 kDa) serving as internal control. (**C**) Cell viability was assessed using the MTT assay in cells treated with LL-37 (0.1, 1, 4 and 10 µM) for 4 h. (**A**-**C**) Data are presented as mean ± SEM, *n* = 8 (**A**,**B**) and *n* = 6–8 (**C**) in each group. Statistical significance was calculated using a one-way ANOVA, followed by Tukey’s post hoc test. *** represents *p* < 0.001 vs. control. For comparisons indicated by the bars, ** and *** represent *p* < 0.01 and *p* < 0.001, respectively.

**Figure 5 biomedicines-10-00492-f005:**
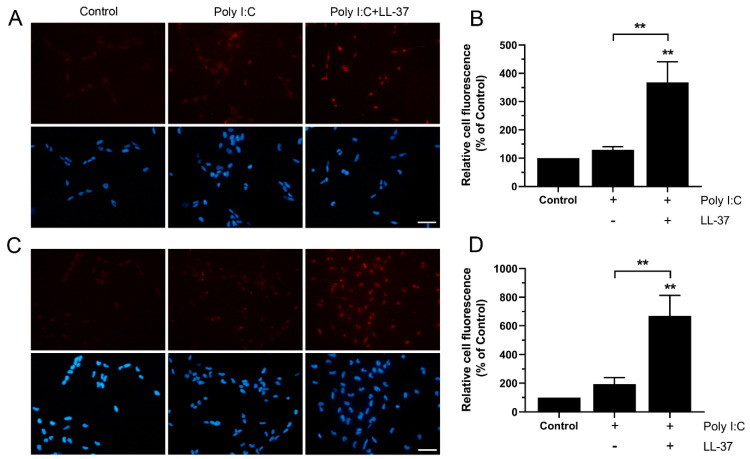
LL-37 triggers import of poly I:C in BEAS-2B cells. (**A**–**D**) Cells were treated with fluorescent rhodamine-tagged poly I:C (4 µg/mL) alone or in combination with LL-37 (4 µM) for 6 (**A**,**B**) and 24 h (**C**,**D**). The intracellular fluorescence signal (red) and the nuclei staining with DAPI (blue) were analyzed and photographed using a fluorescence microscope equipped with a digital camera. The bars in panel **A** and **C** represent 40 µm. The fluorescence intensity of five cells was measured in three different areas on each coverslip, and an average cellular fluorescence intensity was calculated for each experiment. Data are presented as mean ± SEM, *n* = 4 in each group representing the number of independent experiments. Statistical significance was calculated using a one-way ANOVA, followed by Tukey’s post hoc test. ** represents *p* < 0.01 vs. control. For comparisons indicated by the bars, ** represents *p* < 0.01.

## Data Availability

The datasets generated during and/or analyzed during the current study are available from the corresponding author on request.

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
