# Peer review of "LL-37 and Double-Stranded RNA Synergistically Upregulate Bronchial Epithelial TLR3 Involving Enhanced Import of Double-Stranded RNA and Downstream TLR3 Signaling"

_biomedicines, 2022, doi:10.3390/biomedicines10020492_

Round 1

Reviewer 1 Report

Summary: In this study, authors have reported a new mechanism behind upregulation of TLR3 expression in human bronchial epithelial BEAS-2B cell line, which is induced by polycytidylic acid. Authors have provided sufficient evidence to support their observations including usage of endosomal acidification inhibitor chloroquine and nuclear factor kappa B inhibitor dexamethasone. Importantly, they relied on statistical hypothesis testing to validate their hypothesis.

  1. On Page 1, in the provided abstract, authors have listed characterization techniques, such as cell viability assessed by MTT assay etc. Rather, it would be helpful to readers if the actual result is stated as opposed to the characterization technique. Authors may also consider providing full forms of the provided abbreviations, such as TLR3 etc. Over all, the abstract is too technical and authors may consider simplifying it in layman's terms.
  2. On Page 1, in Lines 15 - 18, the following text is not clear, “LL-37 potentiated poly I:C-induced TLR3 mRNA and protein expression and this effect was associated with increased uptake of poly I:C. LL-37/poly I:C-induced up-regulation of TLR3 mRNA expression was prevented by the endosomal acidification inhibitor”. This sentence might be missing a relative clause. In addition, authors may consider simplifying it by splitting it into multiple sentences.
  3. In Figures 1 – 5, authors have chosen 24 h as their long-term cell culture time. What is the rationale behind it and why longer time intervals such as 7 days have not been chosen? It would be more informative if the authors provide this information in the manuscript.
  4. In page 6, in Figure 4C, authors have chosen 4 h for LL-37 treatment and cell viability analysis with MTT assay. Why has not been longer time interval chosen as programmed cell death (apoptosis) may take longer time duration to manifest?

Author Response

Response to reviewer 1

We would like to thank the reviewer for valuable and beneficial suggestions and comments on our manuscript. We have addressed the reviewer’s suggestions as out-lined below.

Summary: In this study, authors have reported a new mechanism behind upregulation of TLR3 expression in human bronchial epithelial BEAS-2B cell line, which is induced by polycytidylic acid. Authors have provided sufficient evidence to support their observations including usage of endosomal acidification inhibitor chloroquine and nuclear factor kappa B inhibitor dexamethasone. Importantly, they relied on statistical hypothesis testing to validate their hypothesis.

1. On Page 1, in the provided abstract, authors have listed characterization techniques, such as cell viability assessed by MTT assay etc. Rather, it would be helpful to readers if the actual result is stated as opposed to the characterization technique. Authors may also consider providing full forms of the provided abbreviations, such as TLR3 etc. Over all, the abstract is too technical and authors may consider simplifying it in layman's terms.

Authors’ response: In the revised version of the manuscript, the “Abstract” has been revised in accordance with reviewer’s suggestions. Please, see page 1 in the revised manuscript.

2. On Page 1, in Lines 15 - 18, the following text is not clear, “LL-37 potentiated poly I:C-induced TLR3 mRNA and protein expression and this effect was associated with increased uptake of poly I:C. LL-37/poly I:C-induced up-regulation of TLR3 mRNA expression was prevented by the endosomal acidification inhibitor”. This sentence might be missing a relative clause. In addition, authors may consider simplifying it by splitting it into multiple sentences.

Authors’ response: Corrected as suggested by the reviewer. Please, see “Abstract”, page 1 in the revised version of the manuscript. 

3. In Figures 1 – 5, authors have chosen 24 h as their long-term cell culture time. What is the rationale behind it and why longer time intervals such as 7 days have not been chosen? It would be more informative if the authors provide this information in the manuscript.

Authors’ response: We have previously demonstrated that the combined treatment with LL-37 and poly I:C for 24 h triggers production of pro-inflammatory cytokines IL-6 and MCP-1 in human coronary artery smooth muscle cells, suggesting that stimulation with LL-37 and poly I:C in combination is pro-inflammatory at this time point (Dahl et al. Inflamm Res 2020;69:579-588). Hence, it is reasonable to assume that stimulation with LL-37/poly I:C also will have an impact TLR3 expression within 24 h. In the revised version of the manuscript, we clarify the rationale for assessing TLR3 transcript and protein expression at 24 h as suggested by the reviewer. Please, see “Results”, page 4, first paragraph in the revised manuscript.       

4. In page 6, in Figure 4C, authors have chosen 4 h for LL-37 treatment and cell viability analysis with MTT assay. Why has not been longer time interval chosen as programmed cell death (apoptosis) may take longer time duration to manifest?

Authors’ response: We completely agree with the reviewer that reduced cell metabolic activity, monitored with the MTT assay, occurs much earlier than apoptosis. High concentrations of LL-37 (>4 µM) cause a rapid (within minutes) permeabilization of the plasma membrane observed by measuring increased release of lactate dehydrogenase (LDH), and this effect is accompanied by reduced cell viability assessed with the MTT method before apoptosis can be observed (Svensson et al. Biochem J 2016;473:87-98; Bankell et al. Peptides 2021;135:170432). It seems that LL-37-induced permeabilization precedes caspase-independent apoptosis which occurs later (Bankell et al. 2021). In the revised version of the manuscript, we explain the rationale behind assessing LL-37-induced cytotoxicity with the MTT method. Please, see “Results”, page 6 in the revised manuscript.

Reviewer 2 Report

Bodahl et al have provided evidence that LL-35 enhances expression of TLR3 by increasing import of poly I:C into the endosome, where inhibition of acidification blocks TLR3 signaling and subsequent upregulation of TLR3 itself. This is a new finding.

Major concerns:

  1. The authors need to provide evidence that IL-7-medi8ated import of poly I:C is directly linked to upregulation of TLR3. Is inhibition of endocytosis of poly I:C related to increased TLR2 signaling after treatment with a high concentration of LL-37?

  1. The effect of dexamethasome on TLR3 expression induced by poly I:C seems to be not associated with signaling triggered by a low dose of LL-37 such as upregulation of IκBα and phosphorylation of p65 and p105 (Figures 1 and 2). Since a high dose of LL-37 enforces TLR3 signaling by poly I:C, it is needed to test whether a high dose of LL-37 affects as upregulation of IκBα and phosphorylation of p65 and p105.

Author Response

Reviewer 2

We would like to thank the reviewer for valuable and beneficial suggestions and comments on our manuscript. We have addressed the reviewer’s suggestions as out-lined below.

Bodahl et al have provided evidence that LL-35 enhances expression of TLR3 by increasing import of poly I:C into the endosome, where inhibition of acidification blocks TLR3 signaling and subsequent upregulation of TLR3 itself. This is a new finding.

 Major concerns:

1. The authors need to provide evidence that IL-7-medi8ated import of poly I:C is directly linked to upregulation of TLR3. Is inhibition of endocytosis of poly I:C related to increased TLR2 signaling after treatment with a high concentration of LL-37?

Authors’ response: We completely agree with the reviewer that it is both interesting and important to characterize LL-37-stimulated import of poly I:C, and to use this information to determine the causal relationship between poly I:C import and LL-37/poly I:C-induced upregulation of TLR3. We strongly believe that this interesting topic, suggested by the reviewer, warrants a new complete full study. Thank you very much for highlighting this matter.

LL-37-induced internalization of poly I:C may involve clathrin-dependent endocytosis but also caveolae/lipid raft-dependent endocytosis represents a putative mechanism. Additionally, poly I:C may diffuse through pores in the plasma membrane caused by LL-37-induced permeabilization. To assess involvement of clathrin- and caveolae/lipid raft-dependent endocytosis, blockers of these processes such as chlorpromazine and filipin, respectively, can be used, but also other inhibitors of endocytosis such as dynasore (clathrin-dependent) and nystatin (caveolae/lipid raft-dependent) and siRNA (clathrin and or caveolin-1 siRNA) are useful. Secondary to characterization of LL-37-stimulated import of poly I:C, functional effects, i.e., LL-37/poly I:C-induced up-regulation of TLR3 should be examined for a complete study. In summary, this issue is a very nice topic for a new complete study.

In the present study, we show association, but not a causal relationship, between LL-37-stimulated poly I:C import and LL-37/poly I:C-induced TLR3 expression, as pointed out by the reviewer. Hence, we have corrected the wording by saying “which may involve”, “may potentiate” and “which may” for clarification in the revised version of the manuscript. Please, see last sentence in “Abstract”, page 1, last sentence in “Introduction”, page 2 and first sentence in the last paragraph of “Discussion”, page 8 in the revised manuscript.      

2. The effect of dexamethasome on TLR3 expression induced by poly I:C seems to be not associated with signaling triggered by a low dose of LL-37 such as upregulation of IκBα and phosphorylation of p65 and p105 (Figures 1 and 2). Since a high dose of LL-37 enforces TLR3 signaling by poly I:C, it is needed to test whether a high dose of LL-37 affects as upregulation of IκBα and phosphorylation of p65 and p105.

Authors’ response: Thank you for highlighting this issue. We agree with the reviewer that treatment with LL-37 and poly I:C in combination has a rather weak effect on NF-κB activity. Importantly, however, our data presented in Figure 2C show that the combined treatment with LL-37 (1 µM) and poly I:C (10 µg/ml) significantly increases the amounts of phosphorylated p105 protein compared to control. Please, note that we have used a high concentration of poly I:C in these experiments. Notably, stimulation with LL-37/poly I:C causes no increase in IκBα levels or decrease in phosphorylated p65 levels, effects which would antagonize the LL-37/poly I:C-induced increase in phosphorylated p105 and NF-κB activity (Figure 2A, B). Thus, considering the summarized effects of LL-37/poly I:C on proteins constituting the NF-κB protein complex, our data strongly suggest that stimulation with LL-37 and poly I:C in combination indeed enhances NF-κB activity. In “Results”, page 5, first paragraph of the revised manuscript, we have included this sentence for clarification: “Considering that stimulation with LL-37/poly I:C increases phosphorylated p105 but has no effect on either IκBα or phosphorylated p65 levels, it is reasonable to conclude that treatment with LL-37 and poly I:C in combination enhances NF-κB activity.”

Round 2

Reviewer 2 Report

Now, this manuscript is acceptable.